# The N-Grams Based Text Similarity Detection Approach Using Self-Organizing Maps and Similarity Measures

**Pavel Stefanovič [1],\*, Olga Kurasova [2] and Rokas Štrimaitis [1]**

[1]  Faculty of Fundamental Science, Vilnius Gediminas Technical University, Saulėtekio al. 11, LT-10223 Vilnius, Lithuania; rokas.strimaitis@vgtu.lt

[2]  Institute of Data Science and Digital Technologies, Vilnius University, Akademijos str. 4, LT–08663 Vilnius, Lithuania; olga.kurasova@mii.vu.lt

\*  Correspondence: pavel.stefanovic@vgtu.lt; Tel.: +370-606-77397

**Abstract:** In the paper the word-level n-grams based approach is proposed to find similarity between texts. The approach is a combination of two separate and independent techniques: self-organizing map (SOM) and text similarity measures. SOM's uniqueness is that the obtained results of data clustering, as well as dimensionality reduction, are presented in a visual form. The four measures have been evaluated: cosine, dice, extended Jaccard's, and overlap. First of all, texts have to be converted to numerical expression. For that purpose, the text has been split into the word-level n-grams and after that, the bag of n-grams has been created. The n-grams' frequencies are calculated and the frequency matrix of dataset is formed. Various filters are used to create a bag of n-grams: stemming algorithms, number and punctuation removers, stop words, etc. All experimental investigation has been made using a corpus of plagiarized short answers dataset.

**Keywords:** self-organizing maps; text mining; text similarity measures; n-grams; frequency matrix

---

## 1. Introduction

Nowadays text mining can be used in different practice areas [1] but the most common are: information extraction and retrieval, text classification and clustering, natural language processing, concept extraction, and web mining. Text analysis can be useful and helps to solve problems, such as plagiarism detection, creating effective anti-spam filters [2], finding duplicates in a large number of documents or finding duplicates on the Internet [3]. Some methods focus on keywords from scientific papers' extraction which helps to find the main aim of papers automatically [4]. In an education system, plagiarism detection is a sensitive issue [5]. Plagiarism is one of the common problems because students keep trying to cheat and present writings which they have not created. Usually the main technique to detect similarity between texts is to extract the bag of words from all text datasets. Then the frequency matrix is created, in other words, texts are converted to numerical expressions. In such a technique, the results depend on selected filters when the bag of words is created. Therefore, it is important to select the right filters to get accurate results. Using this technique, we can analyze all texts or just split it into the parts: sentences, paragraphs, pages or n-grams. Depending on the solving task, n-grams could be formed by using character-level or word-level [2,6]. Similarity results can be evaluated using different methods, for example, statistical, estimation of numerical values, or using various clustering methods, such as k-means, Bayesian, artificial neural networks, etc. [7].

In this paper an approach is proposed to find similarity between texts by integrating not only a numerical estimation but also text clustering and visualization. The text similarity detection is based

on text splitting into word-level n-grams and evaluating it using a self-organizing map (SOM) and four numerical measures. The text analysis using a bag of words is not effective because it is difficult to detect how similar two texts are just analyzing the frequency of separate words. Two different people writing some kind of text individually can use similar or almost the same number of words in the text. The bag of words analysis will show that both texts are similar but there is a bigger probability of accidental text match than analyzing a word-level bag of n-grams [8]. In this paper, four specific measures have been used to evaluate texts similarity: cosine, dice, extended Jaccard's, and overlap coefficient. There is significant different evaluations in the literature [9] but these four measures are commonly used in various fields [10,11]. The other part of the approach is based on the detection of the text similarity of a SOM. The advantage of this method compared with other clustering methods is that we can get a visual representation of all texts in a dataset, cluster, as well as similarities. It helps to make decisions much quicker than analyzing numerical estimation. The main problem of the SOM is that it does not have measures that help to define how similar texts are in the same cell of SOM. Thus, for this reason, it is effective to combine analysis of the texts using SOM and numerical similarity measures. To get accurate results, we extract word-level n-grams of different length from texts and analyze them. It allows us to find the same phrases between different texts. In such a way, instead of a bag of words, we have a bag of n-grams, which characterize all texts. The experimental investigation was made using a corpus of plagiarized short answers dataset.

## 2. Text Similarity Detection

### 2.1. Proposed Approach to Evaluate Text Similarity

As was mentioned earlier, there are various methods to find similarity between texts but mostly in all of them similarity is evaluated by numerical measures and is based on usage of bag of words. Instead of this, we propose an n-grams based approach and result estimating in two ways: visual and numerical. The scheme of the proposed approach is presented in Figure 1. The approach consists of three main parts: text preprocessing, visualization and clustering, and numerical estimation. The main aim of text preprocessing is to find the numerical expression of texts (to find the frequency matrix), which will be used for visualization, clustering, and numerical estimation. The detailed description of text preprocessing is presented in Section 2.2. After the frequency matrix is created, the matrix is given to the SOM, where the dataset is clustered and visualized in the SOM. It allows detecting texts similarity in a visual form. In parallel, the four similarity measures are calculated (numerical estimation). The combination of these two separate techniques allows performing deeper texts similarity analysis. The SOM helps to see the whole text dataset similarity in one map and the numerical estimation justifies and specifies the results quantitatively.

### 2.2. Preparation of Frequency Matrix

To analyze texts, it is necessary to convert textual information to numerical expression; the so-called frequency matrix needs to be created. There are many different tools to create it [12–14], but the main steps are usually the same (Figure 2).

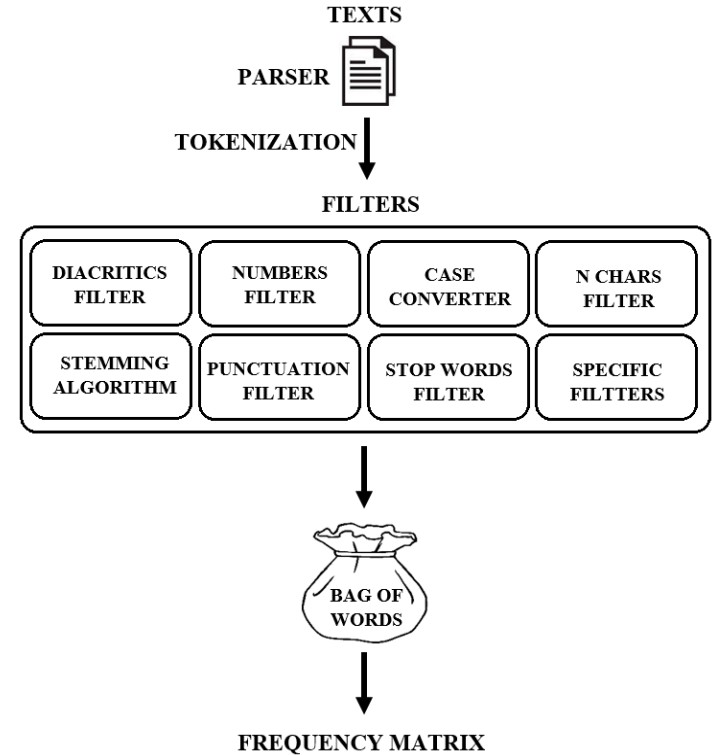

**Figure 1.** The scheme of the proposed approach; SOM: self-organizing maps.

**Figure 2.** The process of creation of frequency matrix.

At first, a text's dataset has to be parsed, all textual information is extracted from the original source and Meta information is not included (pictures, tables frames, schemes, and other not necessary information are rejected). After parsing is done, tokenization has to be made. Tokenization is a process of breaking a stream of text up into words, phrases, symbols, sentences or other meaningful elements

called tokens. The list of tokens becomes input for further processing, such as text mining. Afterwards, we can choose a different filter. It is obvious that all texts have some information that do not characterize the text or is simply not important in the text analysis. Therefore, the aim of selected filters is to reject not important information from texts datasets, such as numbers, punctuation, stop words, etc. The most popular filters and their descriptions are presented in Table 1. In some text mining systems, a specific filter which helps to reject just links, keywords or other not ordinary information can be found. It is obvious that the most important part in text conversation is filter selection because it has the biggest influence on the results. So, it is important to choose the right options, to get accurate results [15]. Otherwise, useful information can be rejected and results can be inappropriate.

**Table 1.** Descriptions of filters.

| Filters | Description |
| --- | --- |
| Diacritics filter | Removes all diacritical marks. Diacritical marks are signs that have been attached to a character usually to indicate distinct sound or special pronunciation. Examples of words (terms) containing diacritical marks are naïve, jäger, réclame, etc. If the specific language texts are analyzed, the diacritical marks cannot be rejected because it can change the meaning of the word. For example, törn/torn (Swedish), sääri/saari (Finnish). |
| Number filter | Filters all terms that consist of digits, including decimal separators ',' or '.' and possible signs '+' or '-'. |
| N chars filter | Filters all terms with less than the specified number N characters. |
| Case converter | Converts all words to lower or upper case. |
| Punctuation filter | Removes all punctuation characters of terms. |
| Stop words filter | Removes all words which are contained in the specified stop word list. Often words such as 'there', 'where', 'that', 'when', etc. compose the stop word list. Not all of them are important for texts analysis. However, the common word list can depend on the domain of texts. For example, if we analyze scientific papers, the words such as 'describe', 'present', 'new', 'propose', 'method', etc. also do not characterize the papers and it is not purposeful to include the words into the texts dictionary. Stop words list can be adapted for any language. |
| Stemming algorithm | The stemming algorithm separates the stem from the word [16]. For example, we have four words 'accepted', 'acceptation', 'acceptance', and 'acceptably'. The stem of the words is 'accept', so only this word will be analyzed, other words are ignored. |

According to the selected filters, a so-called bag of words is created. The bag of words is a list of terms from texts excluding the words that do not satisfy the conditions defined by the selected filters. Suppose we have a texts dataset $D = \{D_1, D_2, \ldots, D_N\}$. According to the frequency of the words in the texts, a so-called frequency matrix is created:

$$\begin{pmatrix} x_{11} & x_{12} & \ldots & x_{1m} \\ x_{21} & x_{22} & \ldots & x_{2m} \\ \vdots & \vdots & \ddots & \vdots \\ x_{N1} & x_{N2} & \ldots & x_{Nm} \end{pmatrix} \tag{1}$$

Here $x_{pl}$ is the frequency of the $l$th word in the $p$th text, $p = 1, \ldots, N$, $l = 1, \ldots, m$. $N$ is the number of the analyzed texts, and $n$ is the number of words in the bag of words. In the simplest case, frequency value is equal to number that shows how many words appear in the text. A row of matrix (1) is a vector, corresponding to a text. The vectors $X_1, X_2, \ldots, X_N$, $X_p = (x_{p1}, x_{p2}, \ldots, x_{pm})$, $p = 1, \ldots, N$, can be used for a text analysis using various methods.

Sometimes it is not enough to analyze just words extracted from texts, especially when similarity has to be found. The analysis of n-grams can be used [17]. An n-gram is a contiguous sequence of $n$ items from a given sequence. The item can be described as word, letter, phonemes, etc. In our research, we have used a word as the item. In this way, we have a bag of n-grams, where each text is characterized by unique n-grams (a few words from the texts). The n-grams analysis allows to

compare a few words in texts, and so obtained similarity results are more accurate. The main steps of n-grams usage are the same as presented in the scheme in Figure 2. We further suggest adding sorting (Figure 1) which helps to avoid a problem when the words in different texts are written in a different order. Suppose we have two n-grams 'data mining methods' and 'methods of data mining'. After filtering (common words are rejected) and the sorting (ascending) step is completed, we get the same n-gram: 'data methods mining'. In the final results, we get the frequency matrix (1) where each $x_{pl}$ will be equal to frequency of the $l$th n-gram in the $p$th text. The proposed approach to find similarity between texts can be used to detect plagiarism (see Figure 1).

### 2.3. Self-Organizing Maps

There are many different clustering methods which can be used in text analysis [18–20]: artificial neural network (ANN), k-means, agglomerative hierarchical clustering, etc. The SOM is one of the most popular artificial neural network models, proposed by Professor T. Kohonen [21]. New extensions and modifications are developed constantly. SOMs can be used to cluster, classify, and visualize the data. The main advantage of this method is to show results in visual form [22]. There are many different tasks where SOM can be used and solve it. SOM can be useful in text mining, too [23,24]. The main aim of SOM is to preserve the topology of multidimensional data when they are transformed into a lower dimensional space (usually two-dimensional). The SOM is a set of nodes, connected to one another via a rectangular or hexagonal topology. The rectangular topology of SOM is presented in Figure 3.

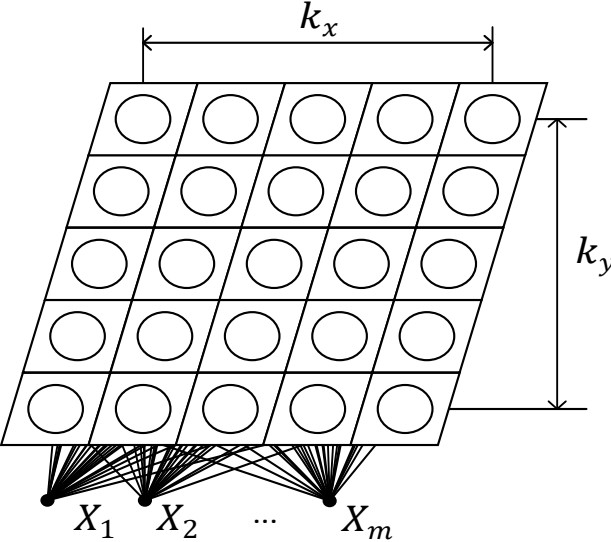

**Figure 3.** Two-dimensional self-organizing map (SOM) (rectangular topology).

The set of weights forms a vector $M_{ij}, i = 1, \ldots, k_x, \ j = 1, \ldots, k_y$ that is usually called a neuron or codebook vector, where $k_x$ is the number of rows, and $k_y$ is the number of columns of the SOM. All texts of the analyzed dataset converted to SOM are given as a matrix (1). The learning process of the SOM algorithm starts from initialization of the components of the vectors (neurons) $M_{ij}$. They can be initialized at random (usually these values are random numbers from the interval (0, 1)) or by the principal components. At each learning step, an input vector $X_p$ is passed to the SOM. The vector $X_p$ is compared to all neurons $M_{ij}$. Usually, the Euclidean distance between this input vector $X_p$ and each neuron $M_{ij}$ are calculated. The vector (neuron) $M_w$ with the minimal Euclidean distance to $X_p$ is designated as a neuron winner (the best match unit). All the neuron's components are adapted according to the learning rule:

$$M_{ij}(t+1) = M_{ij}(t) + h_{ij}^w \big( X_p - M_{ij}(t) \big) \tag{2}$$

Here $t$ is the number of learning step, $h_{ij}^{w}$ is a neighboring function, $w$ is a pair of indices of the neuron winner of vector $X_p$. The learning is repeated until the maximum number of learning step $T$ is reached.

### 2.4. Measures for Text Similarity Detection

To evaluate the similarity between texts, it is necessary to use some mathematical expressions which can evaluate and give the answer to one single numeric value [25,26]. The widest known and used texts similarity measures are cosine, dice, the extended Jaccard's, and the overlap coefficient:

$$\cos(D_1, D_2) = \frac{D_1 \times D_2}{\sqrt{|D_1|} \times \sqrt{|D_2|}} \tag{3}$$

$$\text{dice}(D_1, D_2) = 2\frac{D_1 \times D_2}{|D_1| + |D_2|} \tag{4}$$

$$\text{jaccard}(D_1, D_2) = \frac{D_1 \times D_2}{|D_1| + |D_2| - D_1 \times D_2} \tag{5}$$

$$\text{overlap}(D_1, D_2) = \frac{D_1 \times D_2}{\min(|D_1|, |D_2|)} \tag{6}$$

Here $|D_N| = \left(x_{N1}^2 + x_{N2}^2 + x_{N3}^2 + \ldots + x_{Nm}^2\right)$, $D_1 \times D_2 = x_{11}x_{21} + x_{12}x_{22} + \ldots + x_{1m}x_{2m}$. To show how these four measures are calculated, a simple example is presented. Let us say we have four texts $D = \{D_1, D_2, D_3, D_4\}$ with few words inside of them (Table 2).

**Table 2.** Text dataset.

| Text Inside | Texts |
|---|---|
| text message | $D_1$ |
| computer science | $D_2$ |
| data mining and text mining | $D_3$ |
| methods of text data mining | $D_4$ |

Let us say we do not use any filters, so the bag of words list contains all terms from texts follows: text, message, computer, science, data, mining, and, methods. According to the frequency of each term, the frequency matrix is obtained (Table 3).

**Table 3.** Frequency matrix.

| *text* | *Message* | *Computer* | *Science* | *Data* | *Mining* | *and* | *Methods* | *of* | |
|---|---|---|---|---|---|---|---|---|---|
| 1 | 1 | 0 | 0 | 0 | 0 | 0 | 0 | 0 | $D_1$ |
| 0 | 0 | 1 | 1 | 0 | 0 | 0 | 0 | 0 | $D_2$ |
| 1 | 0 | 0 | 0 | 1 | 2 | 1 | 0 | 0 | $D_3$ |
| 1 | 0 | 0 | 0 | 1 | 1 | 0 | 1 | 1 | $D_4$ |

After a frequency matrix is obtained, we can calculate the similarity measures. The results of calculated measures as an example are given in Table 2, presented in Table 4.

**Table 4.** Results of similarity measures.

| Cosine Measure | | | |
|---|---|---|---|
| | $D_1$ | $D_2$ | $D_3$ | $D_4$ |
| $D_1$ | 100 | 0 | 26 | 31 |
| $D_2$ | 0 | 100 | 0 | 0 |
| $D_3$ | 26 | 0 | 100 | 67 |
| $D_4$ | 31 | 0 | 67 | 100 |
| **Extended Jaccard's Measure** | | | |
| | $D_1$ | $D_2$ | $D_3$ | $D_4$ |
| $D_1$ | 100 | 0 | 13 | 17 |
| $D_2$ | 0 | 100 | 0 | 0 |
| $D_3$ | 13 | 0 | 100 | 50 |
| $D_4$ | 17 | 0 | 50 | 100 |
| **Dice Measure** | | | |
| | $D_1$ | $D_2$ | $D_3$ | $D_4$ |
| $D_1$ | 100 | 0 | 22 | 29 |
| $D_2$ | 0 | 100 | 0 | 0 |
| $D_3$ | 22 | 0 | 100 | 67 |
| $D_4$ | 29 | 0 | 67 | 100 |
| **Overlap Measure** | | | |
| | $D_1$ | $D_2$ | $D_3$ | $D_4$ |
| $D_1$ | 100 | 0 | 50 | 50 |
| $D_2$ | 0 | 100 | 0 | 0 |
| $D_3$ | 50 | 0 | 100 | 80 |
| $D_4$ | 50 | 0 | 80 | 100 |

As we can see, the results of cosine and dice measures are almost the same. The values of extended Jaccard's are lower compared to the others. The overlap measure shows the highest values, and as there is no difference between overlap ($D_1$, $D_3$) and overlap ($D_1$, $D_4$), it means that these texts are equal in the point of similarity. All measures can be used equally to find similarity between texts, so it is hard to say which one is the most accurate and the deep investigation has to be made.

## 3. Experimental Investigation

### 3.1. Dataset

A corpus of plagiarized short answers [27] has been used for experimental investigation. This dataset is also suitable to find similarity between texts. The corpus consists of one hundred texts: 95 answers provided by the 19 participants and 5 original Wikipedia source articles. The questions for students are given bellow:

$Q_1$—'What is inheritance in object oriented programming?'
$Q_2$—'Explain the PageRank algorithm that is used by the Google search engine'
$Q_3$—'Explain the vector space model for Information Retrieval'
$Q_4$—'Explain Bayes Theorem from probability theory'
$Q_5$—'What is dynamic programming?'

For each question, there are 19 examples of each of the heavy revision, light revision, and near copy evaluation, and 38 non-plagiarized examples written independently from the Wikipedia source (Table 5). The average length of text in the corpus is 208 words and 113 unique tokens. The description of each revision level is given:

- Near copy (cut)—participants were asked to answer the question by simply copying the text from the relevant Wikipedia article.

- Light revision (light)—participants were asked to base their answer on the text found in the Wikipedia article and were, once again, given no instructions about which parts of the article to copy.
- Heavy revision (heavy)—participants were once again asked to base their answer on the relevant Wikipedia article but were instructed to rephrase the text to generate the answer with the same meaning as the source text, but expressed using different words and structure.
- Non-plagiarism (non)—participants were provided with learning materials in the form of either lecture notes or sections from textbooks that could be used to answer the relevant question.

**Table 5.** The level of texts plagiarism.

| Texts ID | Category | | | | |
|----------|----------|----------|----------|----------|----------|
| | $Q_1$ | $Q_2$ | $Q_3$ | $Q_4$ | $Q_5$ |
| $D_1$ | non | cut | light | heavy | non |
| $D_2$ | non | non | cut | light | heavy |
| $D_3$ | heavy | non | non | cut | light |
| $D_4$ | cut | light | heavy | non | non |
| $D_5$ | light | heavy | non | non | cut |
| $D_6$ | non | heavy | light | cut | non |
| $D_7$ | non | non | heavy | light | cut |
| $D_8$ | light | cut | non | non | heavy |
| $D_9$ | non | heavy | light | cut | non |
| $D_{10}$ | non | non | heavy | light | cut |
| $D_{11}$ | cut | non | non | heavy | light |
| $D_{12}$ | heavy | light | cut | non | non |
| $D_{13}$ | non | heavy | light | cut | non |
| $D_{14}$ | non | non | heavy | light | cut |
| $D_{15}$ | cut | non | non | heavy | light |
| $D_{16}$ | non | non | heavy | light | cut |
| $D_{17}$ | cut | non | non | heavy | light |
| $D_{18}$ | light | cut | non | non | heavy |
| $D_{19}$ | heavy | light | cut | non | non |
| $D_{20}$ | Original | Original | Original | Original | Original |

*3.2. Steps of the Experiment*

To find the similarity between the analyzed dataset, the experimental investigation was made in three steps. At the first step, the way to create a bag of n-grams was analyzed. The primary research shows that for this dataset, the maximum words in n-grams can be five, otherwise some data is lost because of short texts. In addition, to create the bag of n-grams, all filters given in Table 1 were included. The focus is given when the words in n-grams are equal from three to five, so in total fifteen variants were analyzed. The size of bag of n-grams is given in Figure 4.

At the second step, four similarity measures (Table 6) were calculated between all twenty texts to detect which texts are similar, to compare it with the given categorical descriptions (Table 5), and to decide which measure gives better results. At the last step, the same dataset has been presented with SOM. In SOM, we can see all twenty texts' similarity at once and according to the obtained results, decide how similar each text is to each other.

**Table 6.** Texts similarity results sorted as Near copy (cut), Light revision (light), Heavy revision (heavy), Non-plagiarism (non) (results are given in percent).

| $Q_1$ | | Cut | | | | Light | | | Heavy | | | Non | | | | | | | | |
|---|---|---|---|---|---|---|---|---|---|---|---|---|---|---|---|---|---|---|---|
| | | $D_4$ | $D_{11}$ | $D_{15}$ | $D_{17}$ | $D_5$ | $D_8$ | $D_{18}$ | $D_3$ | $D_{12}$ | $D_{19}$ | $D_1$ | $D_2$ | $D_6$ | $D_7$ | $D_9$ | $D_{10}$ | $D_{13}$ | $D_{14}$ | $D_{16}$ |
| Cosine | $D_{20}$ | 61 | 51 | 55 | **96** | **94** | 19 | 21 | 7 | 53 | 0 | 1 | 0 | 0 | 5 | 0 | 0 | 4 | 0 | 0 |
| Dice | | 59 | 50 | 44 | **96** | **94** | 19 | 20 | 6 | 53 | 0 | 1 | 0 | 0 | 4 | 0 | 0 | 3 | 0 | 0 |
| Extended Jaccard's | | 42 | 33 | 28 | **92** | **88** | 10 | 11 | 3 | 36 | 0 | 0 | 0 | 0 | 2 | 0 | 0 | 2 | 0 | 0 |
| Overlap | | **80** | 66 | 113 | 99 | 97 | 26 | 30 | 10 | 54 | 0 | 2 | 0 | 0 | 8 | 0 | 0 | 5 | 0 | 0 |

| $Q_2$ | | Cut | | | | Light | | | Heavy | | | Non | | | | | | | | |
|---|---|---|---|---|---|---|---|---|---|---|---|---|---|---|---|---|---|---|---|---|
| | | $D_1$ | $D_8$ | $D_{18}$ | $D_{14}$ | $D_{12}$ | $D_{19}$ | $D_5$ | $D_6$ | $D_9$ | $D_{13}$ | $D_2$ | $D_3$ | $D_7$ | $D_{10}$ | $D_{11}$ | $D_{14}$ | $D_{15}$ | $D_{16}$ | $D_{17}$ |
| Cosine | $D_{20}$ | 64 | 28 | **0** | 16 | 29 | 54 | 25 | 9 | 8 | 7 | 0 | 1 | 0 | 7 | 0 | 1 | 0 | 0 | 0 |
| Dice | | 58 | 27 | **0** | 11 | 23 | 50 | 18 | 8 | 7 | 6 | 0 | 1 | 0 | 6 | 0 | 1 | 0 | 0 | 0 |
| Extended Jaccard's | | 41 | 15 | **0** | 6 | 13 | 33 | 10 | 4 | 4 | 3 | 0 | 0 | 0 | 3 | 0 | 0 | 0 | 0 | 0 |
| Overlap | | **100** | 39 | **0** | 42 | 58 | **81** | 56 | 14 | 12 | 12 | 0 | 1 | 0 | 11 | 0 | 1 | 0 | 0 | 0 |

| $Q_3$ | | Cut | | | Light | | | | Heavy | | | | | Non | | | | | | |
|---|---|---|---|---|---|---|---|---|---|---|---|---|---|---|---|---|---|---|---|---|
| | | $D_2$ | $D_{12}$ | $D_{19}$ | $D_1$ | $D_6$ | $D_9$ | $D_{13}$ | $D_4$ | $D_7$ | $D_{10}$ | $D_{14}$ | $D_{16}$ | $D_3$ | $D_5$ | $D_8$ | $D_{11}$ | $D_{15}$ | $D_{17}$ | $D_{18}$ |
| Cosine | $D_{20}$ | **79** | 0 | 24 | 51 | 18 | **77** | 46 | 36 | 32 | 14 | 17 | 45 | 3 | 3 | 6 | 6 | 4 | 2 | 5 |
| Dice | | **78** | 0 | 22 | 51 | 17 | **77** | 45 | 36 | 32 | 14 | 17 | 44 | 2 | 3 | 6 | 6 | 3 | 2 | 5 |
| Extended Jaccard's | | **64** | 0 | 12 | 34 | 10 | **63** | 29 | 22 | 19 | 8 | 9 | 28 | 1 | 1 | 3 | 3 | 2 | 1 | 3 |
| Overlap | | **92** | 0 | 36 | 54 | 20 | **79** | 57 | 39 | 33 | 15 | 17 | 51 | 3 | 4 | 8 | 6 | 6 | 2 | 5 |

| $Q_4$ | | Cut | | | | Light | | | | | Heavy | | | | Non | | | | | |
|---|---|---|---|---|---|---|---|---|---|---|---|---|---|---|---|---|---|---|---|---|
| | | $D_3$ | $D_6$ | $D_9$ | $D_{13}$ | $D_2$ | $D_7$ | $D_{10}$ | $D_{14}$ | $D_{16}$ | $D_1$ | $D_{11}$ | $D_{15}$ | $D_{17}$ | $D_4$ | $D_5$ | $D_8$ | $D_{12}$ | $D_{18}$ | $D_{19}$ |
| Cosine | $D_{20}$ | 60 | 36 | 36 | **98** | 24 | 13 | 59 | 18 | 43 | 14 | 27 | 6 | **93** | 0 | 0 | 0 | 0 | 1 | 0 |
| Dice | | 52 | 36 | 36 | **98** | 21 | 13 | 58 | 17 | 42 | 13 | 27 | 4 | **93** | 0 | 0 | 0 | 0 | 1 | 0 |
| Extended Jaccard's | | 35 | 22 | 22 | **97** | 12 | 7 | 41 | 9 | 27 | 7 | 15 | 2 | **87** | 0 | 0 | 0 | 0 | 0 | 0 |
| Overlap | | **105** | 39 | 39 | 99 | 39 | 17 | 66 | 25 | 56 | 22 | 28 | 16 | **96** | 0 | 0 | 0 | 0 | 1 | 0 |

| $Q_5$ | | Cut | | | | | Light | | | | Heavy | | | | Non | | | | | |
|---|---|---|---|---|---|---|---|---|---|---|---|---|---|---|---|---|---|---|---|---|
| | | $D_5$ | $D_7$ | $D_{10}$ | $D_{14}$ | $D_{16}$ | $D_3$ | $D_{11}$ | $D_{15}$ | $D_{17}$ | $D_2$ | $D_8$ | $D_{18}$ | $D_{13}$ | $D_1$ | $D_4$ | $D_{16}$ | $D_9$ | $D_{12}$ | $D_{19}$ |
| Cosine | $D_{20}$ | 42 | 36 | **77** | 50 | **82** | 35 | 9 | 21 | 62 | 38 | 3 | 31 | 1 | 0 | 0 | 0 | 0 | 1 | 1 |
| Dice | | 30 | 35 | **75** | 40 | **81** | 28 | 9 | 12 | 58 | 36 | 3 | 28 | 1 | 0 | 0 | 0 | 0 | 1 | 1 |
| Extended Jaccard's | | 17 | 21 | **59** | 25 | **69** | 16 | 5 | 7 | 40 | 22 | 1 | 16 | 0 | 0 | 0 | 0 | 0 | 0 | 0 |
| Overlap | | **100** | 46 | **97** | **100** | **97** | 67 | 13 | 65 | 93 | 54 | 6 | 50 | 1 | 0 | 0 | 0 | 0 | 1 | 1 |

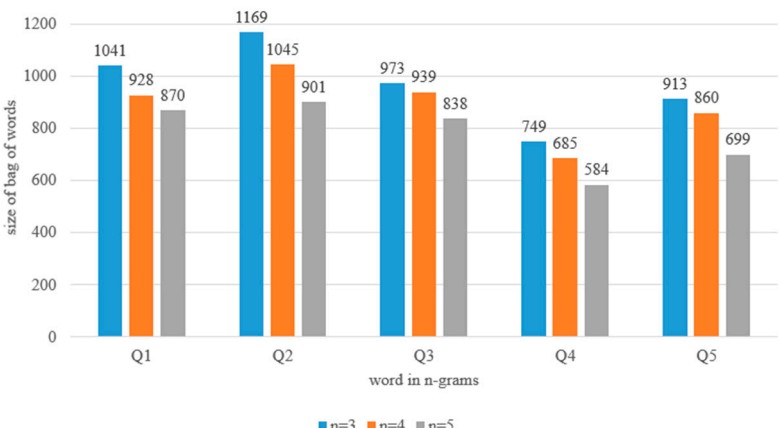

**Figure 4.** The size of the obtained bag of words.

### 3.3. Experimental Results

Deeper analysis showed that for this dataset there was no big difference between three, four or five words n-grams used so the final experimental results will be presented when the bag of n-grams is created using three words. In Table 6, we can see all calculated measures, which represent the similarity between text $D_{20}$ (original text) and other texts in the dataset. The variable $Q_n$, where $n = 1, \ldots, 5$ is the question number from the original dataset [28]. All values in Table 6 are in percent so the lowest percent means the worst result (texts are not similar), the highest-best (similar). The highest percent has been marked in bold.

The highest percent in $Q_1$ analysis were obtained for all measures when $D_{20}$ was compared to texts $D_5$ and $D_{17}$. According to the Table 5, the most similar texts to the original are texts $D_4$, $D_{11}$, $D_{15}$, $D_{17}$. All measures get the highest percent when the original text was compared to the $D_{17}$, which proves that this text is a near copy. The text $D_5$ is marked as light revision, but all measures showed that it is mostly copied text. As we can see, the other near copy texts ($D_4$, $D_{11}$, $D_{15}$) were detected (the highest percent) as a near copy when overlap measure (5) was used alone ($D_4 = 80\%$, $D_{11} = 66\%$, $D_{15} = 113\%$). The value of $D_{15}$ is higher than 100%, because the original text and $D_{15}$ fully overlap and some n-grams in original text were even mentioned a few times more. In this case, it meant that these two texts were totally similar. If we look to the $Q_2$ answers' text similarity results, we can see that the highest percent is obtained with text $D_1$ and $D_{19}$ using the overlap measure. According to Table 5, the heavy copies are $D_1$, $D_8$, and $D_{18}$. So the only one overlap measure can confirm just one near copy text similarity ($D_1 = 100\%$) and light revision ($D_{19} = 81\%$). Neither measure detected the similarity of text $D_{18}$, all of them got 0%. Deeper analysis showed that it was some a mistake given in dataset description because looking at the $D_{18}$ text and comparing it with the original text it was confirmed that these two texts cannot be marked as a near copy because the text is totally different.

The results of questions $Q_1$ and $Q_2$ answers' text similarity are presented using SOM (Figure 5) [24]. The color scale from white to black in cells means the values of the U-matrix [22]. The lighter color means that the distance between some data is short and the dark otherwise. The pie charts represent the texts of the dataset. If the dataset items are very similar among each other, they will fall to the same cell (one pie chart divided to pieces). As we can see in Figure 5a, the texts $D_5, D_{17}, D_{20}$ fall in the same cell so it means that these two texts are similar to the original text $D_{20}$, that was earlier proved by calculating similarity measures. The other texts also make some groups or fall to the same cell. For example, according to the Table 5, the texts $D_2, D_6, D_{10}, D_{16}$ are non-copies so in SOM their fall out in the same cell. Using SOM, we can easily identify which texts are similar to each other. In the right side of Figure 5b, the near copy and light revision texts $D_1, D_4, D_5, D_8, D_{12}, D_{19}$ are located near original text $D_{20}$. It also confirms that these texts are the most similar to the original text.

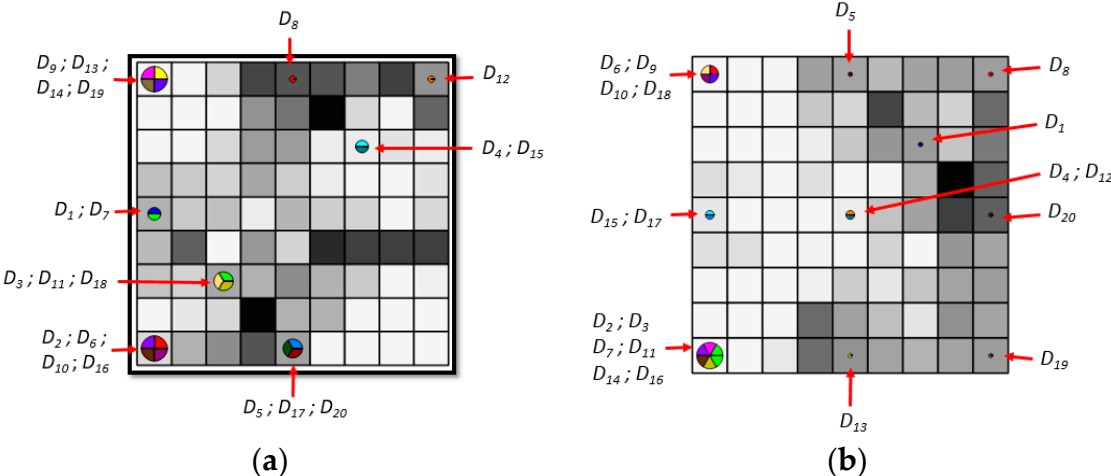

**Figure 5.** $9 \times 9$ self-organizing maps: (**a**) first question ($Q_1$); (**b**) second question ($Q_2$).

The results of $Q_3$ answers texts showed that the highest percent according to the all measures were obtained when the original text was compared to the texts $D_2$ (near copy) and $D_9$ (light revision). The same results as the previous question were found; the highest percent when the overlap measure was used ($D_2 = 92\%$, $D_9 = 79\%$). As we can see, in the right top corner of the SOM (Figure 6a) all these three texts are located in the same cell. The other texts also make some clusters which are formed according to the Table 5 described categories. The results of $Q_4$ answers texts in the SOM (Figure 6b) shows that the texts $D_{13}$ (near copy), $D_{17}$ (light revision) are similar to the original text $D_{20}$, but the other near copy texts $D_3, D_6, D_9$ are located and grouped far away from original text cell. So in this case, the SOM recognize similarity partly. The similarity measures also confirmed similarity of three texts (Table 6) only, where overlap measure gave the highest values: $D_3 = 105\%$, $D_{13} = 99\%$, $D_{19} = 96\%$. In this case the text $D_3$ fully overlapped the original text so it is obviously plagiarism.

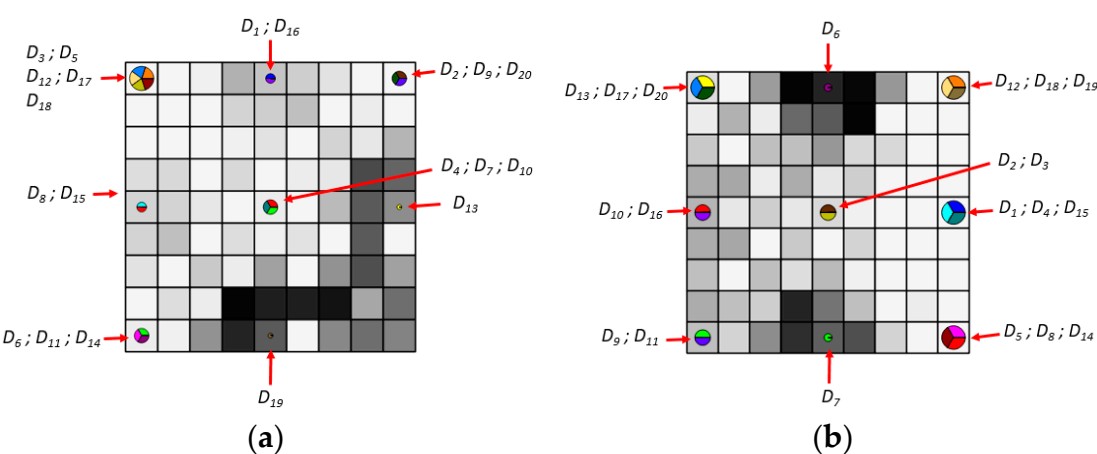

**Figure 6.** $9 \times 9$ self-organizing maps: (**a**) third question ($Q_3$); (**b**) fourth question ($Q_4$).

The results of the last question $Q_5$ answers texts similarity were almost all confirmed by calculated measures. According to Table 5, the near copy texts are $D_5, D_7, D_{10}, D_{14}$, and $D_{16}$. All four measures proved that four of five texts are similar to the original text. As with previous results, the highest percent was obtained using the overlap measure: $D_5 = 100\%$, $D_{10} = 97\%$, $D_{14} = 100\%$, and $D_{16} = 97\%$. Only one overlap measure confirmed the similarity of the text $D_5$, with other measures the value is small. In the bottom left corner of the SOM (Figure 7), the original text $D_{20}$ is located in the same cell with text $D_{16}$. The other near copy texts are scattered over all map so in this case, it is hard to confirm similarity just using the SOM.

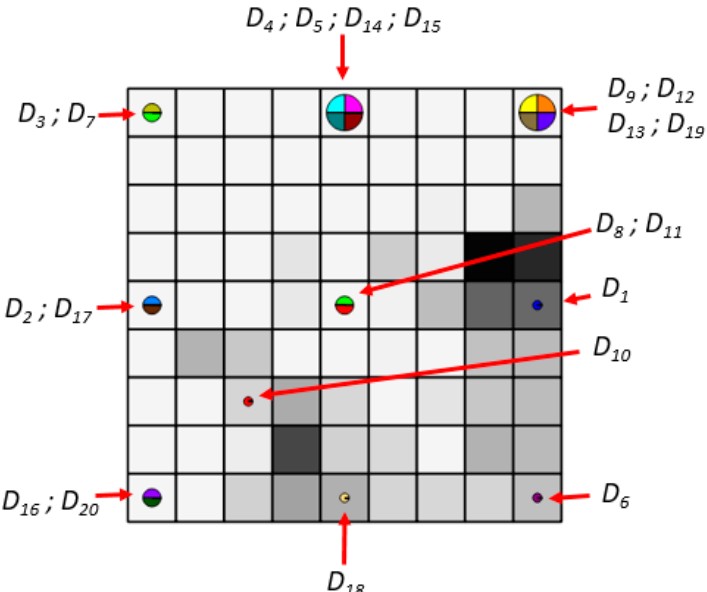

**Figure 7.** $9 \times 9$ self-organizing map: fifth question ($Q_5$).

## 4. Conclusions

In this paper an approach was proposed to detect similarity between texts. The approach was based on the text splitting into n-grams and evaluating it using a SOM and similarity measures. The detection of similar texts was made in three steps: (1) text dataset conversion to numerical expression using n-grams; (2) calculation of similarity measures; (3) text dataset visualization using SOM and similarity representation on it. At the first step, the main focus was to create a bag of n-grams of all datasets. The various number of words in n-grams were analyzed. In addition, different filters were applied: numbers and punctuation removing, words frequency, uppercase transform, stemming algorithm, etc. The analysis showed the filters and size of n-grams influenced the final results. For this dataset, the size of the n-grams was selected and equal to three for the experimental investigation. At the second step, the four similarity measures were calculated: cosine, dice, extended Jaccard's, and overlap. Final results showed that the highest percent of similarity was obtained using overlap measures. The other three measure values were always similar and smaller. The usage of SOM showed that SOM helps to see the summarized results of all texts' similarity in visual form quickly. It is very easy to understand which texts are similar to each other or not. In the analyzed dataset case, the SOM helped to detect similarity, and the formed clusters were correlated with the given categorical description of the dataset.

The experimental investigation showed that the most accurate similarity measure is overlap because this measure detected more near copy texts and gained the highest percent. Sometimes it showed even full texts overlap which can be defined as plagiarism. The SOM helps to summarize the full dataset similarity in visual form, but it is hard to confirm how much texts are similar to each other. The investigations showed that SOM was more useful as an additional tool to decide which texts could be similar and deeper investigation could then be applied. The usage of n-grams and creation of a bag of words showed that it is an effective way to find similarity between texts. Deeper analysis has to be made to detect how all filters, size of n-grams, and other texts' conversation to numerical expression affect the final results for much longer texts' datasets. So it is purposeful to analyze them in more detailed in the future. The proposed approach allowed finding similarity between texts and evaluating results by combining SOM and numerical estimations helped to make a deep analysis.

**Author Contributions:** Designing and performing measurements, P.S., O.K. and R.S.; Data analysis, P.S., O.K. and R.S.; Scientific discussions, P.S., O.K. and R.S.; Writing the article, P.S., O.K. and R.S.

**Funding:** This research received no external funding.

**Conflicts of Interest:** The authors declare no conflict of interest.

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
