# Peer review of "The N-Grams Based Text Similarity Detection Approach Using Self-Organizing Maps and Similarity Measures"

_applsci, doi:10.3390/app9091870_

Round 1

Reviewer 1 Report

The work is interesting, well-motivated and supported by an appropriate number of relevant references. General structure of the paper is adequate. However, I have some minor comments in order to improve the manuscript:

·          Quality of English is not good, so it should be reviewed. Furthermore, I have found errors associated with spelling and grammar. Some examples:

o    Line 110, “frequency value IS equals”

o    Line 110, “appear”, no “appears”

o    Line 116, “have used”, no “have been used”.

o    Line 120 “texts”, no “text”.

o    Line 212 revise the sentence “the creation of bag of n-grams…”

o    Revise the expression “the all”.

o    Line 226 “have”, no “has”.

o    Line 235, measures “show”, no “shows”

o    Line 304, “clusters have been correlated with …”

·          Line 73, must be “preprocessing”, no “reprocessing”

·          In the document must be “text dataset”, no “texts dataset”

·          Line 109, “m” is the number of words in the bag of words, no “n”.

·          Line 139, “Mij” is a matrix.

·          Line 211 Rephrase.

·          Improve the description of Table 4.

Author Response

Thank you for your review. The paper was one more time reviewed, and all errors have been corrected. Some parts of the paper have been rewritten.

Reviewer 2 Report

The paper describes a method for measuring similarity of text documents. Plagiarism was simulated by human essay writing experiments to evaluate the proposed measure. The test setup is interesting and the measure has potential usefulness. The novelty seems to be to use bag-of n-grams at word level. The bag-of-items itself is well-known, and the n-grams has been used at the character-level for text similarity measure, and also word level for other text analysis applications. To get accepted, the paper should be improved in several ways.

Abstract and Introduction needs to be completely re-written. Currently they suffer the following issues:

- What is the current state-of-the-art?
- What are the new contributions in the methodology compared to state-of-the-art?
- Give proper overall system description of the proposed measure.
- Structure the intro properly; now there are 2 over-lengthy paragraphs w/o good structure.
- At the end of intro you mention "bag of n-grams" but never define that it operates with words, not characters. This is essential information for the reader.
- Refer to existing works (a) state-of-the-art in text similarity measures, (b) using n-grams at char level, (c) at word level (if any).

Writing style needs improvements throughout. There are huge problems in writing style that are not language related. Very often text is not logical and lacks fluency. There is many times text that means nothing while the important matters are left unexplained.

- Abstract: To evaluate similarity measure... four similarity measures are evaluated. This is empty sentence that makes no sense. Another one in intro: "similarity detection for text similarity detection". It is like saying "fish is fish". Please remove all empty sentences, they degrade the quality of the paper.
- "Focus is given" ... what does this mean?
- What means "text similarity detection"? I think you introduce a *measure*, not a detector. A detector gives True/False answer, whereas a measure gives a value 0..1. This repeats several times in the text.

Why so much focus on SOM?

a. SOM is basically clustering algorithm. Clustering is made based on similarity measure so that more similar items (text documents) are in the same clusters. Experimenting the goodness of a similarity measure in clustering task is just fine if you have some expected behavior of the clustering, and analyze the results from the expected point of view. However, you report merely that SOM confirms the similarity of texts. But this is not true. The text items go to the same cluster merely because the measure tells they are similar. The result of clustering is therefore merely an application of the measure.
b. SOM is not particularly good clustering algorithm. Several clustering results are not even able to be confirmed by the basic assumptions like cut texts are more similar to original than light, for example.

Existing works

a. Existing state-of-the-art is never revealed, and the proposed measure is never compared even a single other measure. I strongly suggest to study some alternatives and consider adding comparison to char-level n-grams, or some other text similarity measure. The following recent paper [1] has 143 measures implemented and tested (API available on internet). Other bag-or-words approaches are also included - though the proposed word-level n-grams are not (which indicate the proposed measure may be novel, which should be clearly stated if it is new or not).
b. Although those measures in [1] are mainly for short text, some of them might apply for longer texts as well. The proposed measure (3-gram of words) might not apply to short text but this would be worth to test briefly using some of those test sets in [1] and commented.

Minor errors
- "reprocessing"
- "integration ensures the comprehensive evaluation" --- means what?
- Fig.2 has block "ETC". What kind of component is that? In general, I strongly advice not to use the word "etc" anywhere as it really contains no information.
- Table 1: Removing dots from German and Scandinavic words like "jäger" can change the meaning completely. Examples: "törn-torn" (Swedish), "sääri-saari" (Finnish). It might be acceptable to allow such minor degradation for the sake of improving the measure but such limitation should be mentioned.
- Table 1: Lower or Upper case? Although it does not matter for the result, you still need to tell the reader which one you are using.
- Example in Fig.5 includes two stop words (and, of) despite the methodology is claimed to included stop word removal.

Experiments
a. The experimental setup using the human simulated exam answer is interesting. This seems to be taken from ref. [24]. Is this data publicly available for others to re-produce the results? If yes, give link. It is quite important others can repeat the tests, and use the same material.
b. Most essential results are in Table 4 but this is very difficult to follow. The problem is the inconsistency of the labels that seems to follow numbering of the humans rather than the task. Since the identity of the person is never used in the methodology, I strongly suggest to re-label and re-organize the data so that instead of D1-D19, they would be sorted according to the expected similarity: cut - light - heavy - non. Relabeling e.g. A1, A2, A3, A4, A5 for non-plagiarism, B1, B2, B3, B4, B5 for heave, and so on. Then sorting the results in Table 4 accordingly. This would make to follow the results much easier.
c. After sorting, you might compare the RANKING of the results by the measures are they consistent with the expected ranking. This would allow easier comparison of the four studied base level measures.
d. Any comparative method would put the results into context. Either using same base-level similarity from [1], or some complete system for long texts. Now we have no idea are the results better or not than existing measures.
e. SOM results are more or less entertaining. I would simply just list the CLUSTERS in a table. The visualization does not reveal any new insight. Also some better clustering algorithm, even some k-means variant, might just be more consistent here.

References:
[1] Gali et al, Framework for Syntactic String Similarity Measures, ESWA 2019.

Some other titles that might contain related material. Please check:
- DalGTM at SemEval-2016 Task 1: Importance-Aware Compositional Approach to  
Short Text Similarity
- A Compressed Sensing View of Unsupervised Text Embeddings, Bag-of-n-Grams,
and LSTMs
- Word n-gram attention models for sentence similarity and inference
- Text Similarity Measure based on Continuous Sequences
- WORDS VS. CHARACTER N-GRAMS FOR ANTI-SPAM FILTERING
- https://www.ngrams.info/

Author Response

Thanks you for your review, ideas for future works, and valuable comments.

Round 2

Reviewer 2 Report

The paper still does not describe clearly tell what is the current state-of-the-art. Comparisons were also avoided based on argument that their implementation is too difficult to adopt such changes. I think these deficiencies degrade the quality of the paper but I am not insisting on it as the main results in Table 4 are still interesting even as such.

However, the link to the data must be included in the paper. It is bad practice to explain only to the reviewer but omit the change from the manuscript. Please add the link and explain the data with few words to help others to use it. You owe this to the authors who made their data available for others - this time you benefit from it. So share their contribution.

"https://ir.shef.ac.uk/cloughie/resources/plagiarism_corpus.html It has been described inside [23]."

Author Response

Thank you for review, I add the link, comments are in the file.
